# Mechanical Behaviors of Microwave-Assisted Pyrolysis Recycled Carbon Fiber-Reinforced Concrete with Early-Strength Cement

**DOI:** 10.3390/ma16041507

**Published:** 2023-02-10

**Authors:** Yeou-Fong Li, Jie-You Li, Jin-Yuan Syu, Tzu-Hsien Yang, Shu-Mei Chang, Ming-Yuan Shen

**Affiliations:** 1Department of Civil Engineering, National Taipei University of Technology, 1, Sec. 3, Chung-Hsiao E. Rd., Taipei 10608, Taiwan; 2Department of Materials Science and Engineering, National Taiwan University, 1, Sec. 4, Roosevelt Rd., Taipei 10617, Taiwan; 3Department of Molecular Science and Engineering, National Taipei University of Technology, Taipei 10608, Taiwan; 4Department of Mechanical Engineering, National Chin-Yi University of Technology, Taichung 41170, Taiwan

**Keywords:** recycled carbon fiber, fiber reinforced concrete, early-strength concrete, microwave-assisted pyrolysis, splitting tensile strength, impact energy

## Abstract

This study aimed to investigate the mechanical performance of early-strength carbon fiber-reinforced concrete (ECFRC) by incorporating original carbon fiber (OCF), recycled carbon fiber (RCF), and sizing-removed carbon fiber (SCF). Compressive, flexural, and splitting tensile strength were tested under three fiber-to-cement weight ratios (5‰, 10‰, and 15‰). The RCF was produced from waste bicycle parts made of carbon fiber-reinforced polymer (CFRP) through microwave-assisted pyrolysis (MAP). The sizing-removed fiber was obtained through a heat-treatment method applied to the OCF. The results of scanning electron microscopy (SEM) analysis with energy dispersive X-ray spectrometry (EDS) indicated the successful removal of sizing and impurities from the surface of the RCF and SCF. The mechanical test results showed that ECFRC with a 10‰ fiber-to-cement weight ratio of carbon fiber had the greatest improvement in its mechanical strengths. Moreover, the ECFRC with 10‰ RCF exhibited higher compressive, flexural, and splitting tensile strength than that of benchmark specimen by 14.2%, 56.5%, and 22.5%, respectively. The ECFRC specimens with a 10‰ fiber-to-cement weight ratio were used to analyze their impact resistance under various impact energies in the impact test. At 50 joules of impact energy, the impact number of the ECFRC with SCF was over 23 times that of the benchmark specimen (early-strength concrete without fiber) and was also greater than that of ECFRC with OCF and RCF.

## 1. Introduction

Greenhouse gas emissions from cement production and civil engineering activities accounted for 39% of the global energy-related CO_2_ equivalent emissions in 2018 [1]. Reducing carbon emissions from cement production or increasing the service life of concrete structures are two ways to reduce the carbon footprint of cement usage. The use of recycled materials can reduce carbon emissions by reducing energy consumption [2]. Effective and efficient repairs are crucial in extending the service life of concrete structures. In scenarios like damaged highway pavements, bridge decks, or airport runways, where construction time is often limited, repair materials must rapidly restore concrete strength in the early stage, which is one advantage of using high-early-strength concrete. High-early-strength concrete shortens curing time while achieving the required concrete strength, making it a suitable material for repair work [3,4,5,6,7].

In previous research, various cement binders and accelerating admixtures have been successfully developed that enhanced compressive and flexural strength [8,9,10,11,12]. However, some cement binders and accelerating admixtures are more prone to creating early-age cracks due to autogenous and thermal shrinkage resulting from rapid heat release and early age hydration reactions [13,14]. Additionally, high-early-strength concrete is a brittle material, and its use is restricted by its limitations, as its tensile strength is much lower than its compressive strength [15,16,17].

Fiber-reinforced concrete (FRC) can enhance the flexural strength and toughness of concrete, and reduce the effects of shrinkage. Chopped steel, glass, carbon, aramid, basalt, and polypropylene fibers are commonly used in FRC to improve the mechanical performance of concrete [18,19,20,21,22,23,24,25,26,27]. Carbon fiber is one of the most useful materials for reinforcing concrete, particularly in bending and tensile strength [28]. However, using new carbon fiber as reinforcement in FRC is often not economically feasible, as the cost of new carbon fiber is typically high. According to the “Cradle-to-Cradle” concept, waste can become a new resource for the economy or nature, and recycled chopped carbon fiber appears to be a suitable option for FRC [29]. Compared with other heat treatment methods, microwave offers the benefits of using less energy to achieve fast heat transfer, contactless and selective heating for complex mixtures, and better energy efficiency [30]. The microwave-assisted pyrolysis (MAP) process can recycle fiber from fiber-reinforced polymer (FRP) [31].

Carbon fiber-reinforced polymer (CFRP) is widely used in automobiles, bicycles, spaceships, aircraft, wind turbine blades, and sporting goods due to its low density and high strength [32,33,34,35]. Thus, it can be reasonably anticipated that a large amount of CFRP waste needs to be addressed [36]. In line with the concept from the previous paragraph, CFRP waste can be recycled and become a valuable material [37]. However, CFRP composite materials are not easily naturally decomposed, and traditional methods of disposing of CFRP waste, including thermal treatment, solution soaking, and physical crushing, are harmful to the environment [38,39], especially for waste thermosetting composites. The method of physically crushing carbon fiber to recycle it also has some drawbacks, including only obtaining chopped fibers and stubborn resin on the fiber surface [29,40].

The key factors affecting the strength of FRC include fiber direction, uniformity of dispersion, fiber length, and amount of additive in the specimen [41,42,43,44,45]. The surface sizing of the carbon fiber also impacts dispersion uniformity. A study showed that pneumatic dispersion technology can improve the compressive strength and impact resistance of FRC [46,47]. The length of chopped fibers affects the mechanical performance of FRC, with longer fibers leading to higher bending strength and impact resistance, while shorter fibers result in greater compressive strength. However, excessive or clumped fibers in concrete can result in poor mixing, low workability, and reduced strength [48,49,50,51].

In this study, recycled carbon fiber was selected and integrated into ECFRC with the aim of prolonging the life cycle of carbon fiber from CFRP waste and enhancing the service life of concrete structures, particularly in time-critical scenarios. Mechanical properties of ECFRC with original carbon fiber (OCF), recycled carbon fiber (RCF), and sizing-removed carbon fiber (SCF) were evaluated to improve the appreciable toughness, fatigue resistance, and impact resistance of the concrete.

## 2. Materials and Methods

The three principal components of the ECFRC in this article were cementitious materials (i.e., early-strength cement), aggregates, and carbon fibers. The three types of carbon fibers used were processed individually using different methods. This section introduces the materials, materials properties, and experimental methods used in this study.

### 2.1. Carbon Fiber

Carbon fiber is characterized by its non-degradable, non-corrosive, fatigue-resistant and high-specific-strength properties, making it a popular choice in industries such as aerospace, energy production, defense and automotive parts. In this study, the carbon fibers were manufactured by Tairylan Division, Formosa Plastics Group; and the chopped carbon fibers were purchased from Sheng Peng Applied Materials Co., Ltd. The chopped carbon fiber was 24 mm in length and there were approximately 12,000 carbon fiber filaments in each bundle of chopped carbon fibers. The carbon fiber exhibited a tensile strength of 4900 MPa, a modulus of elasticity of 250 GPa, an elongation of 2.0%, a density of 1.81 g/cm^3^, and a diameter of 7.0 μm. Figure 1 shows the image of the chopped carbon fiber.

### 2.2. Sizing-Removed Carbon Fiber 

The surface of carbon fiber is coated with an appropriate amount of sizing during the manufacturing process. However, this sizing can result in non-uniform dispersion of carbon fiber in cement, leading to decreased strength of FRC. In this research, the carbon fiber was heated, and the sizing was removed using a PF-40 muffle furnace from Chuan-Hua Precision, located in New Taipei City, Taiwan. The schematic diagram of the carbon fiber sizing removal process is shown in Figure 2.

### 2.3. Recycled Carbon Fiber

In this study, the RCF was produced from a CFRP bicycle frame scrap using microwave-assisted pyrolysis (MAP) technology. The CFRP bicycle frame was made by Giant Manufacturing Co., Ltd. (Taichung, Taiwan), and the RCF was obtained from Thermolysis Co., Ltd. (Kaohsiung, Taiwan). The MAP technology uses uniform microwave heat radiation to heat the CFRP from the inside out. As shown in Figure 3, the schematic depicts the MAP process. This process converts the waste fiber from a discarded CFRP bicycle frame into a reusable fiber through pyrolysis of the resin in an anaerobic environment. The pyrolyzed resin decomposed into gas, tar, carbon residue, and water. These by-products can be then re-purposed as gas and fuel [52,53,54]. The RCF, approximately 24 mm in length, was retrieved from waste CFRP bicycle frames.

Figure 4 depicts the process of converting waste CFRP bicycle frames into RCF using the MAP method. Figure 4a depicts a CFRP bicycle frame scrap before applying the MAP technique. Figure 4b shows the RCF heated with 300–1000 W of power at 2.45 GHz for 2 h. Figure 4c indicates the chopped RCF after the MAP treatment. Figure 4d demonstrates the chopped RCF dispersed by the high-pressure pneumatic dispersion method.

To investigate the tensile strength difference between OCF and RCF treated using MAP technology, a single-filament tensile test was conducted per ASTM D3822-07 [55]. To improve accuracy, thirty individual filaments were selected from each carbon fiber bundle for testing. The single-filament tensile test results showed the average tensile force of the OCF and RCF were 10.96 ± 1.51 gf and 11.55 ± 1.23 gf, respectively. Figure 5 and Figure 6 illustrate the failure force and displacement for carbon fiber filaments for the OCF and RCF, respectively. From the test results, the tensile strength of RCF was similar to that of carbon fiber. This outcome demonstrated that MAP technology was able to effectively recycle and reuse the CFRP waste while preserving its mechanical strength.

### 2.4. SEM-EDS Characterization

The effects of the substance on the surface of the carbon fiber on the strength of fiber-reinforced concrete were examined. Scanning electron microscopy (SEM) coupled with energy dispersive X-ray spectrometry (EDS) was used to analyze the OCF, the SCF, and the RCF processed by MAP. Figure 7 shows the SEM-EDS observation results of the RCF. Figure 7a represents the appearance of a recycled CFRP fragment sample after MAP treatment, and the SEM images and EDS spectra were shown in Figure 7b,c, respectively.

In addition, the SEM-EDS characterization of the OCF and sizing-removed carbon fiber were also observed by a scanning electron microscope (JSM-7610F, JEOL, Tokyo, Japan), and the results are shown in Figure 8 and Figure 9, respectively. The EDS measurements showed that the carbon fiber with heat-treated process had the highest carbon content over 99.9%, followed by RCF after MAP treatment with a carbon content of 99.4%, and the untreated carbon fiber with a carbon content of 99.0%.

### 2.5. Pneumatic Dispersion

This study used the pneumatic dispersion method to distribute the fiber bundles. The pneumatic dispersion device consisted of a cylindrical container with a high-pressure air valve, which was connected to an air compressor, and a sealable plastic lid with an exhaust vent for pressure relief. The chopped carbon fiber bundles were placed in the container with a sealable plastic lid. High-pressure air was pumped into the container to make the carbon fibers collide and then disperse. The carbon fiber was able to achieve more effective spreading by using this pneumatic dispersion device.

### 2.6. Early-Strength Carbon Fiber-Reinforced Concrete (ECFRC)

In this study, high-early-strength cement was used to prepare ECFRC specimens. Early-strength cement has the characteristics of rapid hardening and provides high strength at the early stage of solidification. It is widely used in patching materials or made into early-strength concrete for rapid repair. The composition of early-strength cement was analyzed by X-ray fluorescence, as shown in Table 1.

The water–cement ratio of the concrete was 0.6. The fineness modulus (F.M.) was 5.95, shown in Table 2. The specimens in the study were composed of cement, sand, coarse aggregate, and fine aggregate in a mixing ratio of 1:1.05:0.75:1.5, respectively.

## 3. Experimental Plan and Setup

To investigate the mechanical performance of ECFRC, apart from testing the ECFRC incorporated with RCF, OCF, and SCF, three fiber weight proportions (5‰, 10‰, and 15‰) were tested. This section introduces the definition of each specimen and the mechanical test methods used in this study.

### 3.1. Experimental Program

To study the effects of various types of carbon fibers and different fiber weight proportions on ECFRC, a total of 190 concrete specimens were produced. At least 3 specimens were prepared for each ratio, and the mechanical properties of the concrete were tested after curing for 7 days. The naming convention is described in Table 3, with labels such as C-N05, C-N10, and C-N15 referring to carbon fiber content accounting for 5‰, 10‰, and 15‰ of cement weight, respectively, and the letter “C” indicating the specimen was used in the compressive test. The number of specimens for various mechanical tests are shown in Table 4.

### 3.2. Slump Test

According to ASTM C143/C143M-20 [56], 25 blows were given to each of the three layers with a tamping rod in the slump cone during the concrete filling. Then, the slump cone was lifted vertically after the concrete cast was completed, and then the slump value of the mixture was measured.

### 3.3. Compressive Test

The compressive test was conducted according to ASTM C39/C39M-01 [57] to evaluate the compressive performance of each benchmark and ECFRC specimen under static loading. The concrete cylindrical specimens had a diameter of 10 cm and a height of 20 cm, and were tested using a universal testing machine (HT-9501 Hong-Ta) located in Taipei, Taiwan.

### 3.4. Flexural Test

As per ASTM C293-02 [58], the flexural performance of concrete, in which specimen had dimensions of 28 × 7 × 7 cm, was examined by the universal testing machine (HT-9501 Series. Hong-Ta, Taipei, Taiwan) with a load cell (WF 17120, Wykeham Farrance, Milan, Italy), and its loading rate was 0.020 MPa/s.

### 3.5. Splitting Tensile Test

The splitting tensile test was conducted in accordance with ASTM C496/C496M-17 [59]. The splitting tensile properties of the ECFRC and the benchmark specimens were determined using a universal testing machine (HT-9501 Series, Hong-Ta, Taipei, Taiwan). The concrete cylindrical specimens had a diameter of 10 cm and a height of 20 cm.

### 3.6. Impact Test

The impact test was conducted using ACI 544-2R [60] to verify the impact performance of concrete and ECFRC under dynamic loading. The impact machine (SP-006, Sheng Peng, Yunlin, Taiwan) had a movable impactor, and it could also be loaded with additional loadings to evaluate the impact resistance of specimens under different impact energies.

## 4. Results and Discussion

Different types of carbon fiber (OCF, RCF and SCF) were incorporated with early-strength cement in varying fiber weight ratios to prepare the ECFRC specimens. The slump value, compressive strength, flexural strength, splitting tensile strength, and impact resistance of each specimen were evaluated for discussion.

### 4.1. Slump Test

All types of carbon fiber were blended into the dry cement in an equal distribution before being mixed with water. Table 5 shows that varying the fiber weight ratio can affect the slump value of ECFRC. Under the same fiber weight ratio, the slump values for the three carbon fibers were similar due to the similar lengths of the fibers and equal distribution achieved through pneumatic blending. Among the different fiber weight ratios, the ECFRC mixture with 5‰ fiber weight had the best workability, resulting in the least concrete segregation, with a slump value of approximately 150 mm. The ECFRC mixture with 15‰ fiber weight had reduced mechanical strength because it was sticky and the cement paste could not completely cover the aggregate, resulting in a slump value of about 35 mm.

### 4.2. Compressive Test

The compressive strength of three types of ECFRC specimens with varying fiber weight ratios, as well as benchmark specimens, were evaluated under uniaxial loading. Figure 10 shows the average compressive strength of the three kinds of ECFRC and benchmark specimens with varying fiber proportions. The ECFRC with 10‰ fiber weight proportion exhibited the highest compressive strength compared with other fiber weight proportions. As seen from Figure 10, the deviations in the compressive strength of the C-N specimens in were the largest among all the specimens. This could be explained by the presence of sizing on the surface of the carbon fiber, which made it more difficult to distribute evenly.

The properties of fiber-reinforced concrete varied significantly based on the type of carbon fiber, despite having the same fiber weight proportion. For instance, the compressive strength of C-W10, C-R10, and C-N10 specimens was 33.18 MPa, 28.25 MPa, and 28.06 MPa, respectively, which was 34.39%, 14.22%, and 13.65% higher than that of specimen C-B. As seen in Figure 10, the compressive strength of specimens with OCF showed greater deviation than the other two types of carbon fiber. This larger deviation can be attributed to the presence of sizing and the orientation of carbon fiber in the ECFRC specimens. Sizing could decrease the binding force between the carbon fiber and cement elements [46], resulting in reduced friction between the two elements and increasing the potential for slippage failure. The orientation of carbon fiber in the specimens cannot be controlled, and in some orientations, the weaker binding force between elements may not effectively maintain the structure of the specimen, causing a larger deviation in compressive strength within the C-N specimens. The C-R and C-W specimens were processed for sizing removal. Without sizing, the carbon fiber and cement elements have more friction, forming a stronger binding force. Although the orientation of the carbon fiber remains uncontrollable, the stronger binding force was able to effectively retain the structure of the specimens during the compressive test, resulting in higher average compressive strength and smaller deviation between specimens. The experimental results revealed that the average compressive strength of ECFRC containing 10‰ fiber weight proportion of carbon fiber was the highest compared with other proportions. For the same fiber weight ratio of 10‰, the C-W10 specimen showed the highest compressive performance, followed by the C-R10 and C-N10 specimens.

### 4.3. Flexural Test

In this subsection, the flexural strengths of ECFRC of RCF, OCF, and SCF with different fiber weight proportions (5‰, 10‰, and 15‰) were tested and compared with the benchmark specimens. Figure 11 shows the average bending strength of the benchmark and the ECFRC specimens with various proportions.

The ECFRC with a fiber weight ratio of 10‰ showed a greater improvement in flex-ural strength than the other ratios. The flexural strength of the F-W10, F-N10, and F-R10 specimens increased by 65.4%, 64.1%, and 56.5%, respectively, when compared with that of the F-B specimens. The results from this subsection conform with the findings from the compressive test: adding carbon fiber can increase the strength of early-strength concrete, and using a 10‰ fiber weight proportion of SCF can further improve its strength.

Figure 12 illustrates the carbon fiber distribution in the concrete under three fiber weight proportions. Figure 12a depicts a concrete specimen with a 5‰ fiber weight proportion, where the amount of carbon fibers is significantly lower than in specimens with other proportions. This insufficient amount of carbon fibers results in a lack of improvement in compressive strength. The concrete specimen with a 10‰ fiber weight proportion is shown in Figure 12b, and it contains a more appropriate amount of carbon fiber, as the carbon fibers are well spread out and not tangled together. Figure 12c shows the carbon fiber distribution of 15‰ fiber weight proportion, where the fibers are tangled together, resulting in the formation of pores that cause cracks and reduce the strength of the 15‰ ECFRC specimens.

### 4.4. Splitting Tensile Test

According to ASTM C496, the splitting tensile strengths of the three types of ECFRC and benchmark specimens were tested. After seven days of curing, specimens were prepared for each fiber weight proportion. Figure 13 shows the average splitting tensile strength of the benchmark and ECFRC specimens with different fiber weight proportions.

When the fiber weight proportion was 5‰, the average splitting tensile strength increased slightly compared with the benchmark. The strength was the highest when the weight proportion was at 10‰, with the average splitting tensile strength of T-N10, T-R10, and T-W10 specimens being 3.39, 3.48, and 3.57 MPa, respectively. However, the strength did not increase with increasing fiber weight proportion; as seen in the specimens with 15‰ fiber weight proportion, the strength decreased to 3.25 MPa, 3.30 MPa, and 3.38 MPa, respectively. T-N10, T-R10, and T-W10 specimens showed the highest increment in strength when compared with that of the benchmark by 19.4%, 22.54%, and 25.7%, respectively. The trend was similar to the results from the compressive and flexural tests. According to the previous test results, the mechanical behavior of the three kinds of ECFRC had a considerable correlation with the amount of carbon fiber added. The EDS data also showed that the carbon content of SCF was over 99.9%, which was higher than other types of carbon fiber. This is consistent with the mechanical behavior test results of ECFRC, with the ECFRC containing SCF showing the best mechanical behavior in this study.

### 4.5. Impact Test

Of the various fiber weight proportions, the ECFRC with a 10‰ fiber weight proportion showed the most improvement in the compressive, flexural, and splitting tensile tests, making it the selected fiber weight proportion for the impact test. The impact resistance of the specimens was influenced by the fiber weight proportion, as seen in Table 6, which shows the average impact number of I-R10, I-N10, and I-W10 under 50 J of impact energy were 338.8, 293, and 352.2, respectively. These values increased by 2312%, 2221%, and 1907%, respectively, compared to the I-B specimen.

The removal of sizing and resin residual on the carbon fiber surface improved the impact resistance of ECFRC. Incorporating carbon fiber into high-early-strength concrete greatly significantly increased the impact number at lower-impact energy, and although the difference was less noticeable at higher-impact energies, it was still consistent. The deviation in each specimen’s value was due to the aggregate distribution beneath the impact area on the specimen’s surface. In the impact test, high impact energy easily caused damage inside the specimens, destroying cracks and fissures that could grow from the beginning of weak points such as the interface of aggregate, cementitious materials, and carbon fiber or beneath the materials with insufficient strength.

According to the SEM analysis and the above mechanical test results, the addition of carbon fiber can enhance the toughness, fatigue resistance, and impact resistance of early-strength concrete by a noticeable increment. However, using fresh carbon fiber may not be the most economical or environmentally friendly option. RCF, obtained from processing waste carbon fiber parts, has the advantage of extending the service life of carbon fiber and reducing energy consumption during production while possessing similar mechanical strength to the OCF. The ECFRC with RCF as reinforcement seems to be an ideal repairing material for concrete pavement, aircraft runway, and highway bridge expansion joint, as it can achieve the required strength in a shorter time while being economical and environmentally friendly.

## 5. Conclusions

Incorporating OCF, RCF, and SCF in high-early-strength concrete, with various fiber weight proportions, can enhance the static and impact mechanical properties of ECFRC. Several conclusions are listed below.
The results of the slump test indicated that the slump value was primarily influenced by varying fiber weight ratios. When using the same ratio, the slump value of concrete made with the three types of carbon fibers was consistent.The early-strength concrete incorporating a 10‰ carbon fiber resulted in higher compressive, flexural, and splitting strength compared with the 5‰ and 15‰ specimens.Under the same fiber weight proportion, the compressive, flexural, and splitting strength of the ECFRC made with SCF was the highest, followed by RCF and then the OCF.The ECFRC with SCF had the highest impact resistance (I-W10), followed by I-R10 and finally I-N10.The ECFRC with 10‰ RCF exhibited higher strengths than the benchmark by 14.2%, 56.5%, and 22.5% in compressive, flexural, and splitting tensile tests, respectively.The carbon contents of the SCF, RCF from waste CFRP bicycle frames with MAP treatment, and OCF were 100%, 99.4%, and 99.0%, respectively. There was a correlation between the mechanical performance of ECFRC and the carbon content of the carbon fiber used: the higher the carbon content, the higher the strength.

## Figures and Tables

**Figure 1 materials-16-01507-f001:**
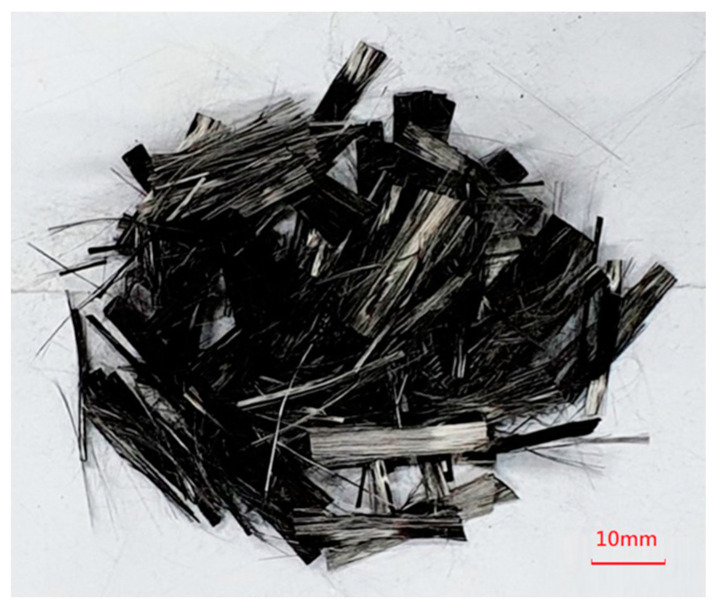
Chopped carbon fiber (24 mm).

**Figure 2 materials-16-01507-f002:**
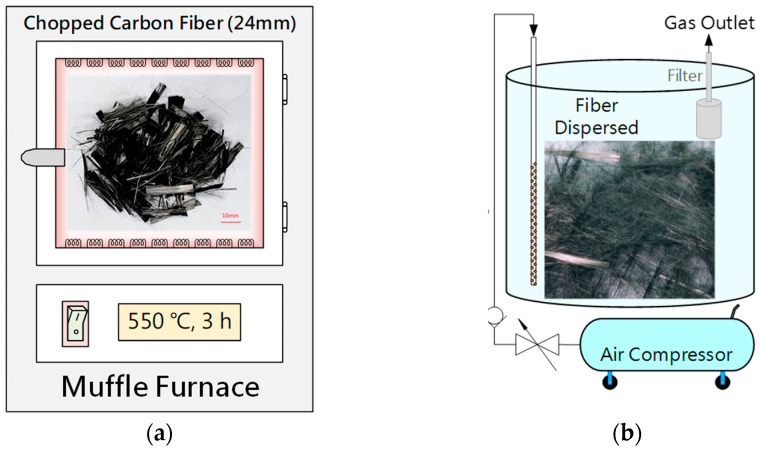
Carbon fiber sizing removal process. (**a**) Heat-treatment method; (**b**) pneumatic dispersion.

**Figure 3 materials-16-01507-f003:**
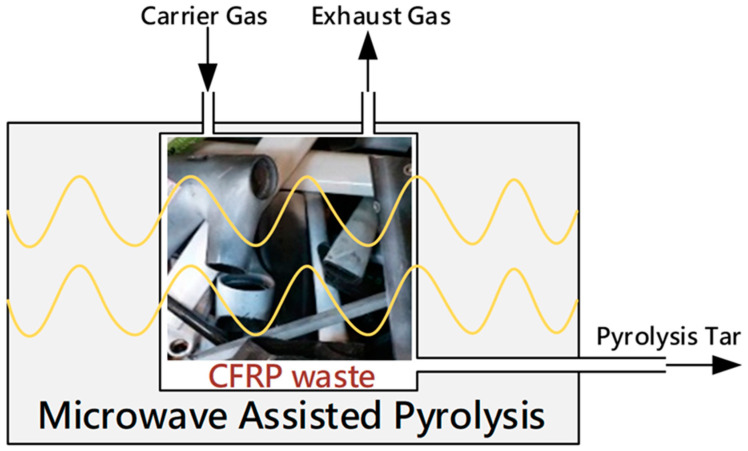
Microwave-assisted pyrolysis instrument.

**Figure 4 materials-16-01507-f004:**
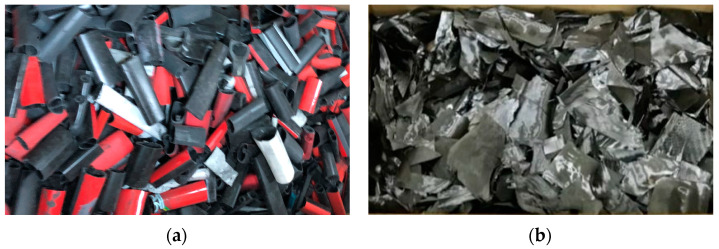
The procedure of carbon fiber recycling. (**a**) CFRP bicycle frame scrap before microwave-assisted pyrolysis; (**b**) microwave-assisted pyrolysis applied specimen; (**c**) recycled carbon fiber by cutting; (**d**) recycled carbon fiber dispersed by pneumatic dispersion.

**Figure 5 materials-16-01507-f005:**
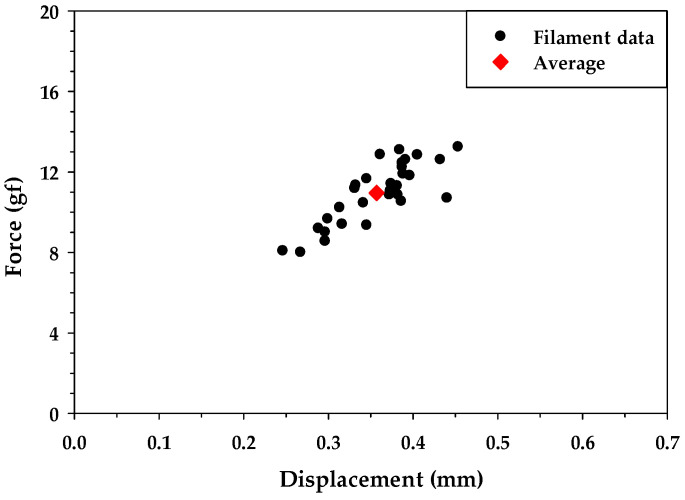
Filament tensile test results of OCF.

**Figure 6 materials-16-01507-f006:**
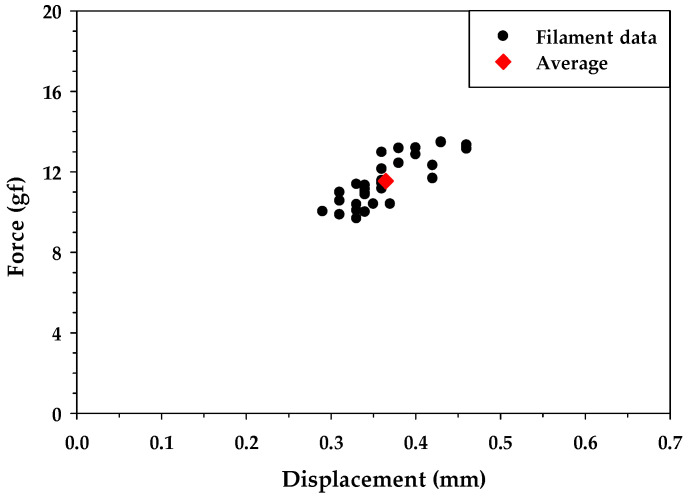
Filament tensile test results of RCF.

**Figure 7 materials-16-01507-f007:**
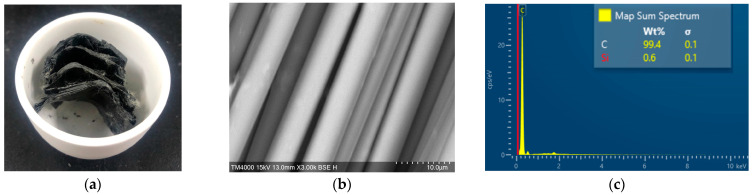
SEM-EDS observation results of the recycled CFRP fragment. (**a**) Image of the CFRP after MAP treatment; (**b**) SEM (3000×); (**c**) EDS.

**Figure 8 materials-16-01507-f008:**
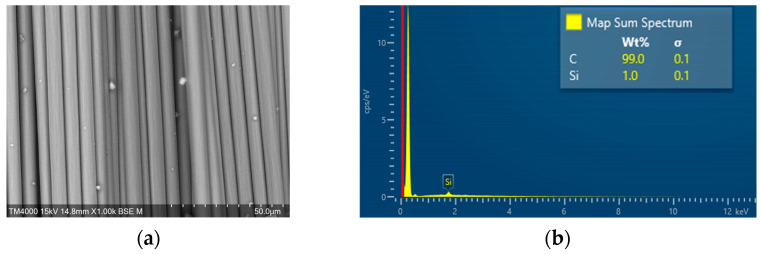
SEM-EDS observation of OCF. (**a**) SEM (1000×); (**b**) EDS.

**Figure 9 materials-16-01507-f009:**
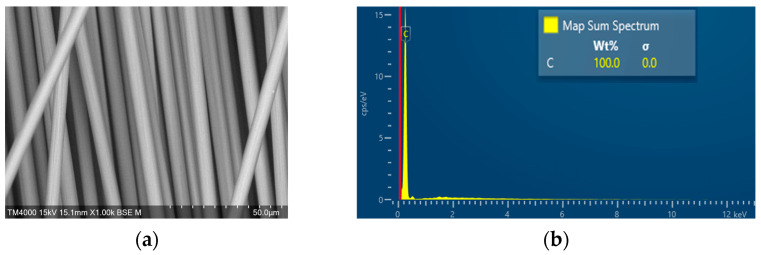
SEM-EDS observation of sizing removed carbon fiber. (**a**) SEM (1000×); (**b**) EDS.

**Figure 10 materials-16-01507-f010:**
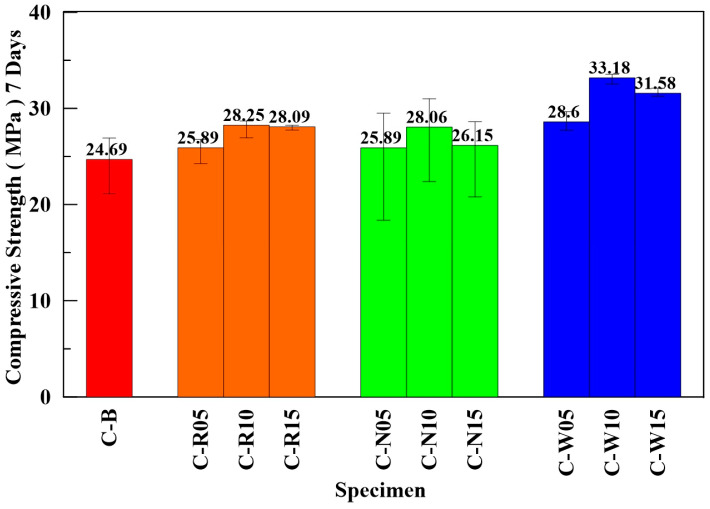
Average compressive strength of ECFRC and benchmark specimens.

**Figure 11 materials-16-01507-f011:**
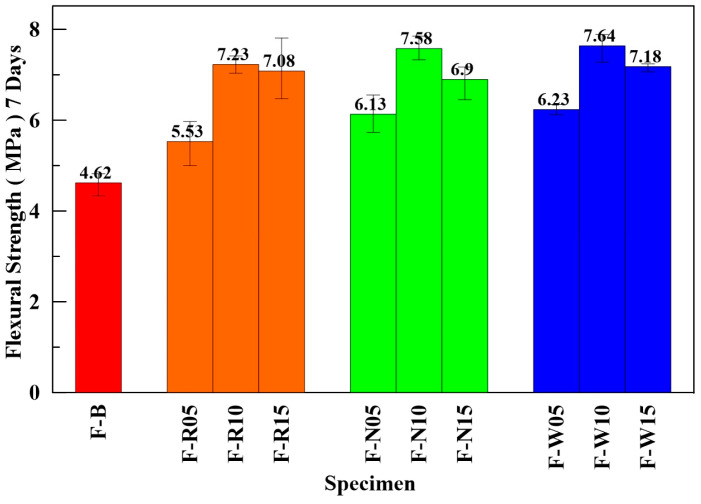
Average flexural strength of ECFRC and benchmark specimens.

**Figure 12 materials-16-01507-f012:**
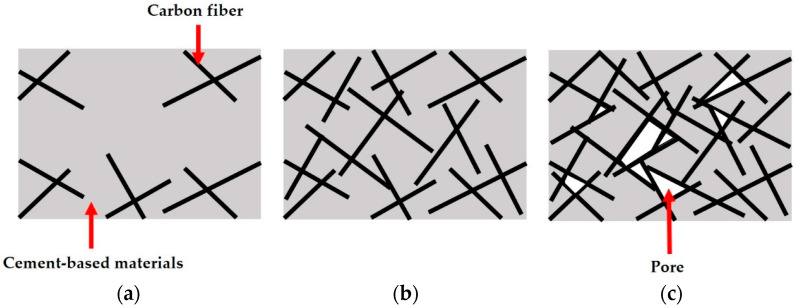
Illustrations of fiber distribution in the concrete specimens. (**a**) Low fiber content; (**b**) optimal fiber content; (**c**) high fiber content.

**Figure 13 materials-16-01507-f013:**
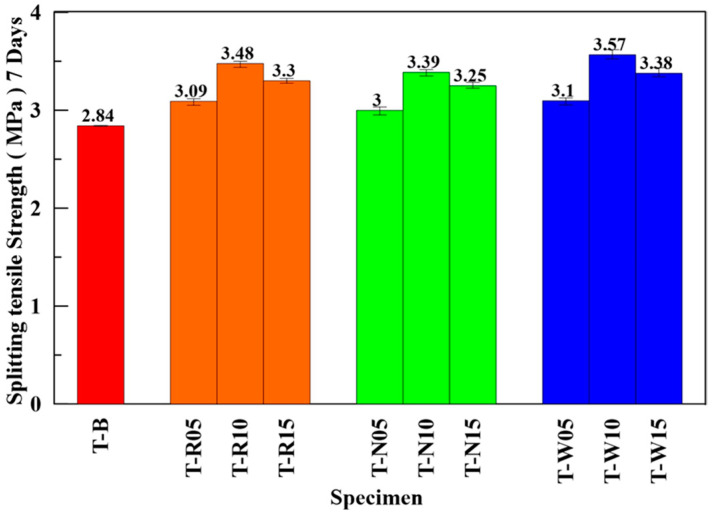
Average splitting tensile strength of ECFRC and benchmark specimens.

**Table 1 materials-16-01507-t001:** Composition of early-strength cement.

Chemical Composition	Percentage (%)
Calcium oxide (CaO)	67.99
Silicon oxide (SiO_2_)	11.83
Aluminum dioxide (Al_2_O_3_)	9.14
Sulfur dioxide (SO_2_)	5.07
Iron (III) oxide (Fe_2_O_3_)	2.88
Potassium oxide (K_2_O)	1.79
Titanium dioxide (TiO_2_)	0.70
Phosphorus pentoxide (P_2_O_5_)	0.50
Other	0.10

**Table 2 materials-16-01507-t002:** Fineness modulus of aggregates.

Sieve No.	Sieve Size (mm)	Weight Retained (g)	Percent Retained (%)	Cumulative Percent Retained (%)
3/2′	37.5	0.0	0.00%	0.00
3/4′	19	723.2	21.92%	21.92
3/8′	9.5	1526.4	46.25%	68.17
No. 4	4.75	14.7	0.45%	68.62
No. 8	2.36	188.0	5.70%	74.31
No. 16	1.18	268.5	8.14%	82.45
No. 30	0.60	209.5	6.35%	88.80
No. 50	0.30	175.6	5.32%	94.12
No. 100	0.15	97.7	2.96%	97.08
Pan	-	96.4	2.92%	-
Total	-	3300.0	100%	595
Fineness modulus (F.M.) = 5.95				

**Table 3 materials-16-01507-t003:** Naming and description of each specimen.

	Naming	Description
Mechanical Testing	C	Compression test
	F	Flexural test
	T	Splitting tensile test
	I	Impact test
ECFRC Specimen	B	Benchmark (without fiber)
	R	Recycled carbon fiber
	N	Original carbon fiber (untreated)
	W	Sizing removed carbon fiber
Fiber Weight Proportion (‰)	05, 10, 15	05 refers to the specimen with a 5‰ fiber-to-cement ratio.

**Table 4 materials-16-01507-t004:** Planning of the number of specimens.

Mechanical Test	Fiber Weight Proportion (‰)	Type of Carbon Fiber			Benchmark (without Fiber)	Total
		OCF	RCF	SCF		
Compression	5	3	3	3	3	30
	10	3	3	3		
	15	3	3	3		
Flexural	5	3	3	3	3	30
	10	3	3	3		
	15	3	3	3		
Splitting Tensile	5	3	3	3	3	30
	10	3	3	3		
	15	3	3	3		
Impact	10	25	25	25	25	100

**Table 5 materials-16-01507-t005:** The slump of ECFRC under different fiber weight proportions.

Addition Proportion of Carbon Fiber (‰)	Recycled Carbon Fiber (R) (mm)	Original Carbon Fiber (N) (mm)	Sizing-Removed Carbon Fiber (W) (mm)
0	230	230	230
5	150	155	155
10	80	80	80
15	35	35	35

**Table 6 materials-16-01507-t006:** Impact test results of benchmark and ECRFC specimens under different impacts.

Specimen	Impact Energy (J)	Specimen Number					Average Impact Number	Increase Percentage (%)
		1	2	3	4	5		
I-B	150	1	1	1	1	1	1.0	-
	125	2	2	2	3	3	2.4	-
	100	3	4	4	5	5	4.2	-
	75	6	6	7	7	8	6.8	-
	50	13	13	14	16	17	14.6	-
I-R10	150	1	2	2	2	2	1.8	80
	125	3	3	4	5	5	4.0	67
	100	14	15	17	20	21	17.4	314
	75	72	75	76	84	86	78.6	1056
	50	320	334	339	347	354	338.8	2221
I-N10	150	1	1	1	2	2	1.4	40
	125	2	3	3	4	5	3.4	42
	100	10	11	14	14	15	12.8	205
	75	59	61	64	68	70	64.4	847
	50	274	281	296	304	310	293	1907
I-W10	150	2	2	2	2	3	2.2	120
	125	3	4	5	5	6	4.6	92
	100	16	18	18	20	22	18.8	348
	75	73	79	88	90	95	85	1150
	50	327	347	355	358	374	352.2	2312

## Data Availability

Data available on request due to privacy.

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
