# Peer review of "Mechanical Behaviors of Microwave-Assisted Pyrolysis Recycled Carbon Fiber-Reinforced Concrete with Early-Strength Cement"

_materials, 2023, doi:10.3390/ma16041507_

Round 1

Reviewer 1 Report

Abstract: provide values for the improvement in properties due to the addition of fibers.

Introduction: clearly identify the research gap at the end of the introduction section and explain clearly the research novelty and significance.

Figure 3 is not clear. please enlarge the picture and include labels to better explain the instrument components.

Figures 5 and 6: provide more details on the test specimen and instrumentation used to obtain these two figures.   

Conclusions: highlight the reduction in slump caused by the addition of fibers. For other conclusions, provide the percent increase in compression, splitting, flexural, and impact strengths quantitatively.  

Reviewer 2 Report

Using the original carbon fiber, recycled carbon fiber, and sizing-removed fiber, the mechanical performances of early-strength carbon fiber-reinforcement concrete were experimentally investigated.  The main parameter is three different fiber-to-cement weight ratios, and the optimal amount of fiber was determined from the test results. Concerning this study, this reviewer has some questions below.

-Please explain why the three different fiber-to-cement weight ratios (5%, 10%, 15%) were set in this study.

-The difference in material characteristics between normal and recycled carbon fibers is important in this study. For the explanation in sections 2.1 and 2.3, please put some tables explaining different mechanical characteristics to show their differences more clearly.

-From Figures 5 and 6, it is said that the tensile strength is similar between normal and recycled ones. About this explanation, the average tensile strength should be indicated. Also, the strength drop after the maximum strength is somewhat unnatural. The tensile strength suddenly drops down in the vertical direction in all specimens. Please explain those behaviors.
